# Unsuccessful treatment outcome and associated risk factors. A prospective study of DR-TB patients from a high burden country, Pakistan

Asif Massud[1,2]*, Amer Hayat Khan[1]*, Syed Azhar Syed Sulaiman[1], Nafees Ahmad[3], Muhammad Shafqat[4], Long Chiau Ming[5]

1 Discipline of Clinical Pharmacy, School of Pharmaceutical Sciences, Universiti Sains Malaysia, Penang, Malaysia, 2 Faculty of Pharmaceutical Sciences, Government College University, Faisalabad, Pakistan, 3 Faculty of Pharmacy, University of Balochistan, Quetta, Pakistan, 4 Programmatic Management of Drug-Resistant Tuberculosis (PMDT) Unit, Nishtar Medical University Hospital, Multan, Pakistan, 5 School of Medical and Life Sciences, Sunway University, Sunway City, Selangor Darul Ehsan, Malaysia

* asifmassud@gmail.com (AM); amer2006@gmail.com (AHK)

**Data Availability Statement:** All relevant data are within the manuscript and its Supporting

## Abstract

### Introduction

Tuberculosis (TB), a curable and preventable infectious disease, becomes difficult to treat if resistance against most effective and tolerable first line anti-TB drugs is developed. The objective of the present study was to evaluate the treatment outcomes and predictors of poor outcomes among drug-resistant tuberculosis (DR-TB) patients treated at a programmatic management unit of drug resistant tuberculosis (PMDT) unit, Punjab, Pakistan.

### Methods

This prospective observational study was conducted at a a PMDT unit in Multan, Punjab, Pakistan. A total of 271 eligible culture positive DR-TB patients enrolled for treatment at the study site between January 2016 and May 2017 were followed till their treatment outcomes were recorded. World Health Organization's (WHO) defined criteria was used for categorizing treatment outcomes. The outcomes of cured and treatment completed were collectively placed as successful outcomes, while death, lost to follow-up (LTFU) and treatment failure were grouped as unsuccessful outcomes. Multivariable binary logistic regression analysis was employed for getting predictors of unsuccessful treatment outcomes. A p-value <0.05 was considered statistically significant.

### Results

Of the 271 DR-TB patients analysed, nearly half (51.3%) were males. The patient's (Mean ± SD) age was 36.75 ± 15.69 years. A total of 69% patients achieved successful outcomes with 185 (68.2%) patients being cured and 2 (0.7%) completed therapy. Of the remaining 84 patients with unsuccessful outcomes, 48 (17.7%) died, 2 (0.7%) were declared treatment failure, 34 (12.5%) were loss to follow up. After adjusting for confounders, patients' age > 50

Information files uploaded along revised submission.

**Funding:** The author(s) received no specific funding for this work

**Competing interests:** The authors have declared that no competing interests exist.

years (OR 2.149 (1.005–4.592) with p-value 0.048 and baseline lung cavitation (OR 7.798 (3.82–15.919) with p-value <0.001 were significantly associated with unsuccessful treatment outcomes.

## Conclusions

The treatment success rate (69%) in the current study participants was below the target set by WHO (>75%). Paying special attention and timely intervention in patients with high risk of unsuccessful treatment outcomes may help in improving treatment outcomes at the study site.

## Introduction

Tuberculosis (TB) is an air borne infectious disease which spreads from person to person and mainly affects lungs though it can also affect other body parts such as spine, kidney, and brains. It is preventable and curable; however, successful TB control becomes difficult if mycobacterium tuberculosis, a pathogen causing TB, become resistant to the most effective and tolerable therapy, thus, the resultant condition termed as drug resistant TB (DR-TB). Availability of fewer controlled trials for evidence of efficacy [1] and limited numbers of drugs for effective management of DR-TB [2], lengthy and tiresome regimens coupled with high cost and toxicity [3, 4], least prioritization to health and lack of political will [5], and deficient health resources required for effective control [6] are some of the major factors that have caused failure to achieve definite treatment outcome goals among DR-TB patients. Even though enormous progress in successful treatment of drug-susceptible TB has been made, Pakistan ranks 4th globally in terms of DR-TB according to a recently published list for high burden countries (HBC) regarding DR-TB patients by World Health Organization (WHO) [7]. An estimated 4.2% (95% confidence interval [CI] = 3.2–5.3%) of new TB cases and 7.3% (95%CI = 6.8–7.8) of previously treated TB cases had DR-TB in Pakistan as per WHO global TB report 2020 [8]. According to global tuberculosis community advisory board, the number of TB patients is increasing rapidly at an estimated rate of 25,000 new cases per year in Pakistan [9]. In 2020, WHO reported that 573,000 TB cases fell ill in Pakistan, out of which 46,000 cases died, while 25000 people were affected by DR-TB [10].

According to WHO global TB report 2021, Pakistan accounts for about 5.8% of new cases globally. In 2009, Pakistan's National Tuberculosis Control Program (NTP) succeeded in obtaining approval form Green Light Committee (GLC) for initiating the use of second line drugs (SLDs) pilot projects for the treatment of 400 DR-TB patients in 3 hospitals. The GLC further approved the required means for the medication management of 1500 DR-TB patients in 2010–2011. Programmatic management of DR–TB (PMDT) in Pakistan was initiated in June 2010 with enrollment of 195 DR-TB for treatment whereas, 2372 DR-TB patients started treatment in 2020 [11]. After PMDT protocol implementation, country has seen considerable progress in terms of DR-TB. Despite being the highest DR-TB burden country in Eastern Mediterranean Region (EMRO), fewer studies have been carried out to assess the treatment outcomes and risk factors associated with unsuccessful treatment outcomes among prospective patients. Majority of already reported studies are retrospective [12–17] and cross sectional [18] in nature thus lacking the coherent study design. We could find only one prospective study among DR-TB patients at some other study site [19] at the time of present study initiation, though 33 PMDT unit are functional to date. To evaluate a healthcare program's efficacy, disease managing protocols and the associated outcomes of a patient cohort should be assessed

[20]. In the absence of prospective studies, retrospective do provide initial point for understanding but lesser control on data and absence of key demographic and clinical parameter, bias in exclusion and inclusion criteria with respect to disease and age and absence of TB drugs information to the patients in these studies raise concerns on the validity of these studies. The reported treatment outcome in these studies cannot be generalized. Smaller sample size, absence of sputum culture data, unavailability of AFB culture facility in some studies, limited demographic knowledge about patients potentiate the need for prospective studies. Due to availability of only retrospective studies with already mentioned deficiencies, present prospective research study was designed to have more control on real and required data. Out-patient DR-TB patients at a high burdened PMDT unit in the densely populated province (Punjab) of Pakistan were evaluated for local drug resistance pattern, therapy outcomes, and the identification of risk factors associated with treatment failure to assess the program effectiveness. The findings of this study would help program coordinators to undertake required measures for the improvement of the TB program. To facilitate the healthcare providers in the management of DR-TB patients, identification of the high-risk patients at an earlier stage, information regarding risk factors for unsuccessful treatment outcome, and drug resistance pattern among local population is extremely helpful.

The resistance to drugs and the outcomes of a treatment regimen are greatly affected by local epidemiology, and if taken into consideration, it helps to devise an optimized empirical therapy. The present study was aimed to assess the pattern of resistance to the treatment regimen and factors associated with unsuccessful treatment outcome among DR-TB patients at the PMDT unit which serve a treatment hub to a densely populated geographic location with no previous studies with the suitability of the current treatment protocols.

## Methodology

### Study population and site

A prospective observational study was carried out at PMDT unit, at pulmonology ward, Nishtar Medical University (NMU), Multan, Punjab, Pakistan. Free of cost necessary diagnostic services are provided to DR-TB patients by pathology and radiology departments of NMU, Hospital. Samples for drug susceptibility testing (DST) are sent to national reference laboratory (NRL), Islamabad, Pakistan. Resistance to rifampicin (R) is considered as pre-requisite for the 18-month DR-TB treatment post sputum culture conversion with second line anti-TB drugs (SLDs). A total of 271 culture confirmed DR-TB patients got enrolled for treatment at the study site between January 2016 and May 2017. Written or oral consent, whichever applicable was obtained from the enrolled patients. All the patients were briefed about the study objectives. The study was approved by the Institutional Ethical Review Board (IRB), NMU, Hospital, Multan, Pakistan.

### Diagnosis and treatment of DR-TB patients

WHO definitions were followed regarding patient identification and diagnosis [21]. Diagnosis and treatment of DR-TB patients at PMDT units in Pakistan have previously been discussed elsewhere [19, 22]. In summary, suspected DR-TB cases, referred to the study site, were initially collected with two sputum samples for sputum microscopy (Zielh-Neelson stain) and Gene Xpert for rifampicin resistance. After obtaining Rifampicin resistance and positive sputum microscopic results, patients were initiated for DR-TB treatment with empirical regimen, except those with previous history of fluoroquinolones, as recommended by national guidelines for DR-TB [22]. Sputum samples for DST result were sent to National reference laboratory (NRL) Pakistan. Patients were documented for any comorbidity before the initiation of

the therapy with the help of their medical record. Enrolled patients were given conventional long regimen treatment (LTR). On the availability of sputum culture and DST results against all first line (FLDs) and second line (SLDs) anti-TB drugs, study participants were shifted to individualized regimen based on patient specific resistance pattern. The aim was to have at least 4 likely effective anti-TB SLDs with maximum recommended daily dose.

Patients, enrolled in study, received medication for a minimum of 18-months after culture conversion. Culture conversion was defined as the consecutive two negative sputum culture results collected at least 30 days apart. Injectable anti-TB drugs were administered for at least 8-months, for a minimum of 6-month post culture conversion during the intensive phase. DR-TB patients were treated as out-patients, and they were assessed on regular monthly interval. Medication adherence was monitored by specially trained support staff. Patients were provided cards, and each dose administered was marked on individual patient card. These cards were counter evaluated on monthly visit by clinician. Treatment compliance was confirmed by a home facilitator. Health facilitator paid home visits and acted as link between patients and PMDT unit treatment staff. Patients were provided free medication for monthly usage. In addition to medication, patients and therapy supporter were entitled to receive transport charges and monthly food ration.

## Data collection

A standardized and comprehensive data collection form was used for patients' socio-demographic, microbiological and clinical data. WHO guidelines defined criteria for management of DR-TB were followed for reporting of treatment outcomes. Cure and treatment completion were grouped under successful/favorable outcomes, whereas death, treatment failure and loss to follow up (LFTU) were categorized as unsuccessful/unfavorable outcomes. Treatment efficacy was calculated by the successful treatment outcome (sum of cured+treatment completed cases) divided by the sum of all cases (cured + treatment completed + died + treatment failure + loss to follow up). Loss to follow up patients have been grouped in "unsuccessful treatment outcome" which is in-line with the WHO and national tuberculosis program (NTP), Pakistan guidelines. Similar grouping has been reported in the published literature as well [23, 24]. As loss to follow up DR-TB patients abort the treatment and hence ultimately impact the overall success rate (cured and completed) of the study and study site performance.

Patients suffering with co-morbidities were recorded for their known diagnosis. Patients were evaluated on monthly basis by disease specialist as per NTP, Pakistan guidelines. Information about demographics and clinical history (age, gender, marital and residential status, smoking, previous TB history, length of disease, previous SLDs use, co-morbidities) and baseline parameters (laboratory, DST result, sputum grading, cavitation) along with monthly clinical data input were recorded. Laboratory tests, conducted on monthly basis, included complete blood count (CBC), serum electrolytes, liver function tests (LFTs), renal function tests (RFTs), random blood glucose and uric acid. Thyroid test, hepatitis, and HIV screening were done at the initiation of therapy. Visual and audiometry tests were done on recommendation of clinician for some patients and were repeated when deemed necessary on physician judgment. All patients were treated free of cost on ambulatory basis with monthly support allowance and transportation charges. Patients clinical record was used for the identification of any co-morbidity. Patients having more than three times levels of the upper value of transaminases or screened confirmed hepatitis (A, B, and C) were defined as hepatitis patient at baseline.

## Statistical analysis

Data was analyzed by statistical package of social sciences version 26 (SPSS Inc., Chicago, IL). Continuous variables were presented as means ± SD (standard deviation), medians and ranges, whereas categorical data was presented as frequencies and percentages. Univariate logistic regression analysis was used to evaluate association between independent variables and unsuccessful treatment outcomes. All variables, considered in univariate logistic regression analysis were based on literature review and suggestions from clinical team at the study site. P-value < 0.05 was used to describe statistical significance of any included variable. Multivariable logistic regression analysis was used to assess the risk factors for unsuccessful treatment outcomes. Relevant independent variables with p-value < 0.2 in univariate logistic regression analysis were included in the multivariable logistic regression analysis [25]. P-value < 0.05 was used to describe statistical significance of any included variable in final analysis.

## Results

### Description of the DR-TB patients

Among 308 enrolled DR-TB patients, 37 patients did not meet the inclusion criteria and were excluded. Pregnant women (5), children age < 18 years (31) and intellectually disable patient (1) were among the excluded from the study. Of the 271 DR-TB patients, 134 (49.5%) patients were only rifampicin resistant and 128 (47.23%) were resistant to both isoniazid and rifampicin (MDR-TB). Both rifampicin resistant (RR) and MDR patients were nearly the 97% of the cohort. Poly drug resistant (PDR) patient (1) included in the study had resistance against rifampicin and pyrazinamide. There were 8 extensively drug resistant (XDR) TB patients accounting to 2.95% of the cohort.

### Patient characteristics

The socio-demographic and baseline clinical characteristics of the 271 DR-TB patients included in the study are shown in Table 1.

### Drug resistance pattern

Drug resistance pattern among all 271 DR-TB patients was documented. Among FLDs, after rifampicin, the rate of resistance was highest for isoniazid (49.4%) followed by pyrazinamide (23.6%), ethambutol (16.6%) and streptomycin (8.1%). Noticeable number of patients were found to be resistant to SLDs (26%). After the availability of DST results for SLDs drug resistance, resistance was highest for *ofloxacin* (Ofx) (24.7%), followed by *kanamycin* (Km) (3%), *amikacin* (Am) (1.8%), *capreomycin* (Cm) (1.5%), and *ethionamide* (Eto) (0.7%). More detailed resistance pattern is given in Table 2.

### Treatment outcomes

Of the 271 patients included in the final analysis, 69% achieved successful treatment outcomes (cured and treatment completed) while unsuccessful treatment outcome included 48 (17.7%) died, 34 (12.5%) loss to follow up, and 2 (0.73%) treatment failure patients (Table 3). Cause of death among DR-TB patients was either TB or clinical conditions due to TB disease progression i.e., cardiac arrest, Myocardial infarction, or chronic illness.

**Table 1. Patients' socio-demographic, baseline clinical characteristics (N = 271).**

| Variable | No. (%) |
|---|---|
| **Gender** | |
| Female | 132 (48.7) |
| Male | 139 (51.3) |
| **Age (Years) (Mean ± SD = 36.75 ± 15.69)** | |
| Age < 50 years | 210 (77.5) |
| Age ≥ 50 years | 61 (22.5) |
| **Marital Status** | |
| Unmarried | 76 (28.0) |
| Married | 195 (72.0) |
| **Residence** | |
| Rural | 131 (48.3) |
| Urban | 140 (51.7) |
| **Employment status** | |
| No | 165 (60.9) |
| Yes | 106 (39.1) |
| **Smoking status** | |
| Non-smoker | 240 (88.6) |
| Active +Ex-smoker | 31 (11.4) |
| **Treatment Registration Category** | |
| New | 38 (14.0) |
| Relapse | 6 (2.2) |
| Treatment after failure | 198 (73.1) |
| Treatment after Loss to follow up | 26 (9.6) |
| Others | 3 (1.1) |
| **Previous TB treatment** | |
| No | 38 (14.0) |
| Yes | 233 (86.0) |
| **Previous use of SLDs** | |
| No | 244 (93.4) |
| Yes | 17 (6.3) |
| **Comorbidity** | |
| No | 226 (83.4) |
| Yes | 45 (16.6) |
| **Patient weight at baseline (Kg) (Mean ± SD = 45.44±11.61)** | |
| < 40 Kg | 188 (69.4) |
| ≥40 Kg | 83 (30.6) |
| **Haemoglobin Level at baseline** | |
| Normal | 82 (30.3) |
| < Normal | 189 (69.7) |
| **Baseline Smear grading** | |
| Neg | 23 (8.5) |
| *Scanty**+1 | 133 (49.1) |
| ***+2‡+3 | 115 (42.4) |
| **Baseline Pulmonary Cavitation** | |
| No Cavitation | 119 (43.9) |
| Cavitation | 152 (56.1) |
| **Resistance to all five FLDs** | |

*(Continued)*

**Table 1.** (Continued)

| Variable | No. (%) |
|---|---|
| No | 257 (94.8) |
| Yes | 14 (5.2) |
| **Resistance to SLDs** | |
| No | 200 (73.8) |
| Yes | 71 (26.2) |

FLDs, first-line anti-TB drugs; SLDs, second line anti-TB drug; TB, tuberculosis

*1–9 Acid Fast Bacilli/100 High Power Field

**10–99 Acid Fast Bacilli/100 High Power Field

*** 1–9 Acid Fast Bacilli/ High Power Field

‡ >9 Acid Fast Bacilli/ High Power Field; Kg, Kilogram; SD, Standard Deviation

**Table 2. Drug resistance pattern of studied patients (N = 271).**

| Resistant Drugs | No. (%) |
|---|---|
| Isoniazid (H) | 134 (49.4) |
| Ethambutol (E) | 45 (16.6) |
| Pyrazinamide (Z) | 64 (23.6) |
| Streptomycin (S) | 22 (8.1) |
| All First Line Drugs (FLDs) | 14 (5.2) |
| Any Second Line Drugs (SLDs) | 71 (26.2) |
| Amikacin (Am) | 5 (1.8) |
| Kanamycin (Km) | 8 (3.0) |
| Capreomycin (Cm) | 4 (1.5) |
| Ofloxacin (Ofx) | 67 (24.7) |
| Ethionamide (Eto) | 2 (0.7) |
| RH | 262(97.0) |
| RH + Ofx | 63 (23.2) |
| HRZ | 63 (23.2) |
| HRZ + Ofx | 39 (14.3) |
| HRE | 44 (16.2) |
| HRE + Ofx | 25 (9.3) |
| HRZE | 29 (10.7) |
| HRZE + Ofx | 19 (7) |
| HRES | 15 (5.5) |
| HRS + Ofx | 13 (4.8) |
| HRZS | 18 (6.6) |
| HRZS + Ofx | 11(4.0) |
| HRES + Ofx | 9 (3.3) |
| HREZ + Km + Am + Ofx | 1 (0.3) |
| HRZ + Km + Am + Ofx | 5 (1.8) |
| All FLDs + Ofx | 9 (3.3) |
| HR + Ofx + Km | 8 (2.9) |
| All FLDs + Ofx + Eto | 1 (0.3) |
| HR + Cm | 4 (1.4) |
| HRS + Ofx + Km | 4 (1.4) |
| HRS + Ofx + Km + Moxifloxacin | 1 (0.3) |

**Table 3. Treatment outcomes of the study participants (N = 271).**

| Treatment outcomes | No. (%) |
|---|---|
| **Successful Treatment outcomes** | **187 (69.0)** |
| Cured | 185 (68.3) |
| Completed | 2 (0.7) |
| **Unsuccessful Treatment Outcome** | **84 (31.0)** |
| Died | 48 (17.7) |
| Failed | 2 (0.7) |
| Loss to follow Up | 34 (12.5) |

## Predictors of unsuccessful treatment outcomes

In univariable logistic regression analysis, the age of participants > 50 OR 2.333 (1.294–4.206), p-value 0.005, married subjects OR 2.008 (1.075–3.752), p-value 0.029, individuals with SLIs resistance OR 4.17 (1.151–19.343), p-value = 0.031), individuals with baseline cavitation (OR 7.147 (3.701–13.804), p-value < 0.001), resistance to all five FLDs OR 0.356 (0.078–1.626) p-value 0.182, Co-morbidity OR 0.587 (0.276–1.25) p-value 0.167 , history of SLDs use OR 2.07 (0.77–5.569) with p-value 0.15, resistance to fluoroquinolones OR 2.067 (1.165–3.670) with p-value 0.013 and 4 or more than 4 resistant drugs OR 1.568 (0.867–2.836) with p-value 0.136 were associated with poor treatment outcome as described in Table 4.

In multivariable logistic regression analysis, after adjusting the marital status, history of SLDs use, co-morbidity, resistance to all FLDs, resistance to fluoroquinolones, resistance to second line injectables (SLIs) and resistance to 4 or more than 4 drugs, significant association was observed between DR-TB patients with age ≥ 50 years OR 2.149 (1.005–4.592) with (p-value 0.048) and baseline lung cavitation OR 7.798 (3.82–15.919) with (p-value <0.001) and unsuccessful treatment outcome as presented in Table 5.

## Discussion

By the end of the study period, treatment outcomes for 271 patients (100%) were available, and 187 (69%) patients achieved treatment success. The study site did not reach the WHO criteria of ≥ 75% target [26]. Treatment success rate of our study was lower comparable to success rates reported elsewhere, [15, 27–29] but better than studies from Bahawalpur Pakistan (59.2%) and 60% [16, 30], China (57%) [31], a meta-analysis [32], meta-analysis [29, 33–35] and GLC-supported DOTS-plus projects [36].

High loss to follow up rate (12.5%) in our study caused decreased success rate. Higher loss to follow up rate in our study could be due to factors such as lack of TB disease knowledge, distance of patients' residency from healthcare settings, fading of symptoms during the early months of anti-TB treatment, age or gender of subjects, and adverse drug effects associated with treatment. Cure rate of our study was better than most of the studies carried elsewhere in world. This could possibly be due to use of individualized regimens and competent people hired by PMDT to provide directly observed treatment (DOT) throughout the treatment duration (18 months) [29, 37, 38]. Additional factors such as age > 65 years, non-alcoholics, and HIV negative status were also observed in our study subjects which are predictive of the success of TB treatment [39]. A mortality rate of 17.7% was observed in our study which was similar to studies reported elsewhere [29, 33, 36, 37] but lower than a study in Peru (53.2%) [40]. A possible reason for the lower mortality rates in the afore mentioned studies [29, 33, 36, 37] might be the masking of deaths by high loss to follow up rates.

**Table 4. Univariable logistic regression analysis of risk factors associated with unsuccessful treatment outcomes among DR-TB patients (N = 271).**

| Variable | Unsuccessful treatment outcome No. (%) | OR (95% CI) | p-value |
|---|---|---|---|
| **Gender** | | | |
| Female | 41 (31.1) | Referent | |
| Male | 43 (30.9) | 0.994 (0.594–1.664) | 0.982 |
| **Age (years)** | | | |
| <50 | 56 (26.7) | Referent | |
| ≥ 50 | 28 (45.9) | 2.333 (1.294–4.206) | **0.005** |
| **Weight** | | | |
| ≥ 40 | 55 (29.3) | Referent | |
| < 40 | 29 (34.9) | 1.299 (0.749–2.251) | 0.352 |
| **Marital status** | | | |
| Unmarried | 16 (21.1) | Referent | |
| Married | 68 (34.9) | 2.008 (1.075–3.752) | **0.029** |
| **Residence** | | | |
| Rural | 42 (32.1) | Referent | |
| Urban | 42 (30.0) | 0.908 (0.543–1.520) | 0.714 |
| **Employment** | | | |
| No | 53 (32.1) | Referent | |
| Yes | 31 (29.2) | 0.873 (0.514–1.485) | 0.617 |
| **Comorbidity** | | | |
| No | 74 (32.7) | Referent | |
| Yes | 10 (22.2) | 0.587 (0.276–1.250) | **0.167** |
| **History of TB Treatment** | | | |
| No | 12 (31.6) | Referent | |
| Yes | 72 (30.9) | 0.969 (0.463–2.027) | 0.933 |
| **History of SLD use** | | | |
| No | 76 (30.0) | Referent | |
| Yes | 8 (47.1) | 2.070 (0.770–5.569) | **0.150** |
| **Smoking** | | | |
| No | 74 (30.7) | Referent | |
| Yes | 10 (33.3) | 1.128 (0.504–2.529) | 0.769 |
| **Baseline sputum grading** | | | |
| Negative | 3(13.0) | Referent | |
| Scanty, +1 | 44 (33.1) | 3.296 (0.929–11.691) | **0.065** |
| +2, +3 | 37 (32.2) | 3.162(0.884–11.317) | **0.077** |
| **Lung cavitation at baseline** | | | |
| No | 13 (10.9) | Referent | |
| Yes | 71 (46.7) | 7.147 (3.701–13.804) | **< 0.001** |
| **Resistance to H** | | | |
| No | 40 (29.2) | Referent | |
| Yes | 44 (32.8) | 1.186 (0.708–1.985) | 0.517 |
| **Resistance to Z** | | | |
| No | 62 (30.0) | Referent | |
| Yes | 22 (34.4) | 1.225 (0.675–2.222) | 0.504 |
| **Resistance to E** | | | |
| No | 73 (32.3) | Referent | |
| Yes | 11 (24.4) | 0.678 (0.325–1.414) | 0.300 |
| **Resistance to S** | | | |

*(Continued)*

**Table 4.** (Continued)

| Variable | Unsuccessful treatment outcome No. (%) | OR (95% CI) | p-value |
|---|---|---|---|
| No | 77 (30.9) | Referent | |
| Yes | 7 (31.8) | 1.042 (0.409–2.659) | 0.931 |
| **Resistance to all 5 FLDs** | | | |
| No | 82 (31.9) | Referent | |
| Yes | 2 (14.3) | 0.356 (0.078–1.626) | **0.182** |
| **Resistance to FQ** | | | |
| No | 55 (27.0) | Referent | |
| Yes | 29 (43.3) | 2.067 (1.165–3.670) | **0.013** |
| **Resistance to any SLI** | | | |
| No | 78 (29.8) | Referent | |
| Yes | 6 (66.7) | 4.178 (1.151–19.343) | **0.031** |
| **Number of resistant drugs** | | | |
| <4 | 60 (28.7) | Referent | |
| ≥ 4 | 24 (38.7) | 1.568 (0.867–2.836) | **0.136** |

OR Odds ratio; CI, Confidence Interval; SLDs, Second line anti-TB drugs; Scanty, 1–9 Acid Fast Bacilli/100 High Power Field; +1, 10–99 Acid Fast Bacilli/100 High Power Field; +2, 1–9 Acid Fast Bacilli/ High Power Field; +3 >9 Acid Fast Bacilli/ High Power Field; FLDs, First line anti-TB drugs; FQs, Fluoroquinolones; SLIs, Second line Injectables.

In multivariable analysis, patients' age ≥ 50 years emerged as a risk factor for death and treatment failure. The younger the research participants, the more likely they are to be healed. Older age is a well-known risk factor for treatment failure in both drug-susceptible and drug-resistant tuberculosis because of these factors. According to the 2014 Global Burden of Disease estimates, the majority of TB-related deaths occurred among the elderly [41]. Age 40 years and above is found to be risk factor for treatment failure in the previous studies as well [19, 42–44]. There is a poor response of older patients towards anti-TB treatment due to general fatigue, co-morbidities, complex medication schedule, poor diet and deficient immunity as stated in studies carried out previously [45–47]. The risk of mortality in DR-TB patients was more than two times greater in older patients, and the risk doubled with every 10 years rise in age [44]. Turkey has documented a similar increased risk of adverse treatment results in elderly DR-TB patients [48]. These factors make older age a risk factor for unfavorable treatment outcomes in patients with DR-TB.

Lung cavitation at baseline was another predictor of poor treatment outcomes in our study which is in line with studies conducted in other parts of world [33, 37, 43, 46, 49]. Patients with lung cavities on their baseline chest X-ray were considerably more likely to have unsatisfactory treatment results in the present group and had more severe and advanced illness and took longer to seek medical help. Cavitary illness is related with a higher degree of infectiousness because to the larger organism burden. Reduced efficacy of antibacterial drugs due to reduced penetration in the presence of lung cavities could be a reason for the poor outcomes in this group of patients [49]. Several studies using qualitative smears and cultures concluded the presence of higher mycobacterial loads in the sputum of patients with cavitary TB [50–52] which could result in a high recurrence rate [53, 54]. Several prior studies have established that cavitary illness is a risk factor for poor treatment results in DR-TB patients, which is consistent with our findings. [37] revealed bilateral lung cavitation as a risk factor for poor treatment results in 15% and 10% of DR-TB patients, respectively, and connected it to mortality and treatment failure. DR-TB patients with bilateral cavitary illness were 2.6 times more likely to

**Table 5. Multivariable logistic regression analysis of risk factors for unsuccessful treatment outcome.**

| Variable | OR 95% CI | p-value |
|---|---|---|
| Age $\geq$ 50 Years | 2.149(1.005–4.592) | **0.048** |
| Marital status | 2.116 (0.985–4.546) | 0.055 |
| Comorbidity | 0.561(0.242–1.3) | 0.178 |
| Previous use of SLDs | 2.484 (0.722–8.548) | 0.149 |
| Baseline Sputum grading (Negative) | Referent | |
| Scanty, +1 | 2.27(0.553–9.32) | 0.255 |
| +2, +3 | 1.653(0.397–6.881) | 0.49 |
| Baseline Lung Cavitation | 7.798(3.82–15.919) | **<0.001** |
| Resistance to All Five FLDs | 0.192(0.03–1.234) | 0.082 |
| Resistance to FQs | 1.174(0.498–2.766) | 0.715 |
| Resistance to SLIs | 4.094(0.645–25.987) | 0.135 |
| Resistance to $\geq$ 4 drugs | 1.612(0.626–4.155) | 0.323 |

B, beta; S.E, Standard error; OR Odds ratio; CI, Confidence Interval; SLDs, Second line anti-TB drugs; Scanty, 1–9 Acid Fast Bacilli/100 High Power Field; +1, 10–99 Acid Fast Bacilli/100 High Power Field; +2, 1–9 Acid Fast Bacilli/ High Power Field; +3 >9 Acid Fast Bacilli/ High Power Field; FLDs, First line anti-TB drugs; FQs, Fluoroquinolones; SLIs, Second line injectables

have unsatisfactory treatment results in a Russian research [33]. Similar findings which suggests that treatment outcome is adversly affected due to presence of lung cavitation at the basline have been reported in another study among DR-TB patients in Pakistani population [19].

Single center study at one of the high burden sites in the geographic area may pose limitation to the study. Although patients of all types of drug resistance were included in the study, yet the generalization of results needs that future research at multicenter PMDT unit sites should be carried out. Patients who died were documented as having died during treatment with the actual reason of death, yet there is need to further evaluate the association of risk factors related to mortality.

## Conclusion

Treatment success rate of our study was not promising as it was low than the WHO global End TB set goal of 75% success rate, thus it needs improvement. The low success rate may be attributed to high loss to follow up rate which needs serious efforts to engage the loss to follow up patients by proper counselling, educating them about their disease, and strategies formulation to enhance patient compliance to therapeutic plan and medication. Rational use of medication may increase the success rate with early detection of resistance pattern and individualized regimen. This study was conducted on subjects from a single center; hence the findings of this study should be confirmed through multi-centered and increased sample size research.

## Supporting information

**S1 Data.**
(XLSX)

## Author Contributions

**Conceptualization:** Asif Massud, Amer Hayat Khan.

**Data curation:** Nafees Ahmad.

**Formal analysis:** Asif Massud, Nafees Ahmad.

**Investigation:** Asif Massud.

**Methodology:** Asif Massud, Muhammad Shafqat.

**Project administration:** Asif Massud.

**Resources:** Muhammad Shafqat.

**Supervision:** Syed Azhar Syed Sulaiman, Nafees Ahmad, Muhammad Shafqat.

**Validation:** Long Chiau Ming.

**Writing – original draft:** Asif Massud.

**Writing – review & editing:** Nafees Ahmad, Long Chiau Ming.

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
