## [Decision Letter · Decision Letter 0]

21 Oct 2022

PONE-D-22-21686Unsuccessful treatment outcomes and associated risk factors: A prospective study of DR-TB patients from a high burden country, PakistanPLOS ONE

Dear Dr. Massud,

Thank you for submitting your manuscript to PLOS ONE. After careful consideration, we feel that it has merit but does not fully meet PLOS ONE’s publication criteria as it currently stands. Therefore, we invite you to submit a revised version of the manuscript that addresses the points raised during the review process.

We look forward to receiving your revised manuscript.

Kind regards,

Ari Samaranayaka, PhD

Academic Editor

PLOS ONE

Journal Requirements:

3. We note you have included a table to which you do not refer in the text of your manuscript. Please ensure that you refer to Table 4 in your text; if accepted, production will need this reference to link the reader to the Table.

Additional Editor Comments:

1. Lost-to-followup (N=34) included in “unsuccessful treatment” group. Need to justify that or need to discuss the possible influence of that in results and conclusions.

2. abstract. “The mean age of patients was 36.75±15.69 years”. Suggest rewording because mean is a single number.

3. Authors say “All relevant data are within the manuscript and its Supporting Information files”. I haven’t seen person level data.

4. line 59-62. I cannot understand the incidence given as a percentage. What is the denominator? Is this the prevalence?

5. I did not find the study aims within introduction section. It is needed to assess the appropriateness of methodology to achieve that aim. What this prospective study aimed to achieved? I had to rely on the brief version found in the abstract.

6. Do you really need figure1? What additional info it provides than those in lines 160-162?

Looks like 271 individuals were resulted from 37 individuals. Need to re-draw Figure 1 to show the flow from 308 to 271 if to be retained.

7. Analyses presented are described multivariate, but they are actually multivariable.

8. Line 111. “any comorbidity”. Did you use a specific definition for comorbidity here?

9. lines 155-157. Two sentences are contradictory to each other. This makes unclear how the variables in table5 were identified. Comparison of tables 4 and 5 tells me only the 1st of these sentences is the correct one. It agrees with the cover letter too. So I wonder about the 2nd sentence.

10. line 164. spellings. “bothe”

11. table1 footnote: remove everything that are not applicable to this table.

12. Why “TB family history” is investigated as a possible factor in table1 when the TB is an infectious disease? By family history do you mean “those living with”, excluding ancestors?

13. line 189. “cured and treatment completed” or “cured or treatment completed”?

14. table3. died=48. does this include deaths due to any reason (eg, road accident) or death due to TB only?

15. Table3. defaulted=34. should this be ‘loss to followup’?

16. table4. this table not cited anywhere in the text. all the tables and figures must be cited even if they contain intermediate results.

17. Table5. B and SE are redundant in the presence of OR, CI, and Pvalue.

18. All available patients were included. But small sample size with only 84 unsuccessful outcomes compared to number of parameters in the mdel presented in table5 is a limitation.

Reviewers' comments:

Reviewer's Responses to Questions

**Comments to the Author**

1. Is the manuscript technically sound, and do the data support the conclusions?

Reviewer #1: Yes

2. Has the statistical analysis been performed appropriately and rigorously? 

Reviewer #1: Yes

3. Have the authors made all data underlying the findings in their manuscript fully available?

Reviewer #1: Yes

4. Is the manuscript presented in an intelligible fashion and written in standard English?

Reviewer #1: Yes

5. Review Comments to the Author

Reviewer #1: This prospective observational study was conducted to evaluate the treatment outcomes and predictors of poor outcomes among drug-resistant tuberculosis (DR-TB) patients treated at a programmatic management (PMDT) unit, in Multan, Punjab, Pakistan. The patients were enrolled for treatment at the study site between January 30, 2016, and May 2017 and were followed till their treatment outcomes were achieved.

The purpose of the study was to evaluate the treatment outcomes and predictors of poor outcomes among drug-resistant tuberculosis (DR-TB) patients.

Dependent (outcome) variables: unsuccessful treatment outcome.

Independent (predictors) variables: gender, age, marital status, residence, employment status, smoking status, treatment registration category, previous TB treatment, previous use of second line drugs, comorbidity, TB family history, patient weight at baseline, haemoglobin level at baseline, baseline smear grading, baseline pulmonary cavitation, resistance to all 5 FLDs, resistance to second line drugs.

A total of 271 eligible culture positive DR-TB patients enrolled for treatment at the study site between January 30, 2016, and May 2017 were followed till their treatment outcomes were recorded. Using the WHO defined criteria, those cured and treatment completed were collectively placed as successful outcomes while those who died, lost to follow-up (LTFU) and failed treatment as unsuccessful outcomes. The data was analyzed using multivariate binary logistic regression and obtained predictors of unsuccessful treatment outcomes. A p-value <0.05 was considered statistically significant. The study was approved by the Institutional Ethical Review Committee (IRB), NMU, Hospital, Multan, Pakistan.

The study findings revealed that of the 271 DR-TB patients analysed, nearly half (51.3%) were males. The mean age of patients was 36.75±15.69 years. A total of 69% patients achieved successful outcomes with 185 (68.2%) patients being cured and 2 (0.7%) completed therapy. Of the remaining 84 patients with unsuccessful outcomes, 48 (17.7%) died, 2 (0.7%) were declared treatment failure, 34 (12.5%) were loss to follow up. After adjusting for confounders, patients’ age > 50 years (OR 2.149 (1.005 41 - 4.592) with p-value 0.048 and baseline lung cavitation (OR 7.798 (3.82 - 15.919) with p-value <0.001 were significantly associated with unsuccessful treatment outcomes.

The authors made all data underlying the findings fully available. However, the data was from only one centre, hence the findings of the study may not be generalized to whole of Pakistan. The authors have however recommended a multi-centered study and/or increased sample size research to address this gap.

The data was also analyzed using both descriptive and inferential statistics which were rigorous and appropriate.

Discussions of the results were robust, citing similar studies conducted both within and outside Ethiopia.

Conclusions are in line with the findings

Writing quality and clarity: Satisfactory

Other observations:

1. Limitations of the study: The authors did well to mention the limitations of the study

2. Inclusion/exclusion criteria clearly explained and demonstrated in a figure form.

6. PLOS authors have the option to publish the peer review history of their article (what does this mean?). If published, this will include your full peer review and any attached files.

Reviewer #1: **Yes: **Haruna Ismaila Adamu, MBBS; MPH; PhD

---

## [Author Response · Author response to Decision Letter 0]

14 Dec 2022

As per journal requirements, supporting data information file has been uploaded as "supporting information" which was lacking at the first manuscript submission.

As suggested by the reviewer, figure has been removed as its was just the duplication of already provided information.

---

## [Editor Report · Decision Letter 1]

21 Dec 2022

PONE-D-22-21686R1Unsuccessful treatment outcomes and associated risk factors: A prospective study of DR-TB patients from a high burden country, PakistanPLOS ONE

Dear Dr. Massud,

Thank you for submitting your manuscript to PLOS ONE. After careful consideration, we feel that it has merit but does not fully meet PLOS ONE’s publication criteria as it currently stands. Therefore, we invite you to submit a revised version of the manuscript that addresses the points raised during the review process.

We look forward to receiving your revised manuscript.

Kind regards,

Ari Samaranayaka, PhD

Academic Editor

PLOS ONE

Journal Requirements:

Additional Editor Comments (if provided):

Most points I raised in the previous review are now addressed successfully in the revised R1 submission. However some of the points are not adequately addressed. I list them below.

• Authors have a good explanation for including lost-to-followup people in the unsuccessful group as explained in the rebuttal letter. This is a context specific explanation, and therefore the same explanation is invalid for most studies in other contexts. Therefore authors’ explanation need to be included in the manuscript for readers.

• I questioned authors’ description of mean age as “36.75±15.69 years”. Their response implies they haven’t understood the point I raised. For example, according to accepted mathematical standards, 10±3 is read as 10-3 or 10+3, meaning 7 or 13. Therefore, reporting mean age as 10±3, need to be read as mean age is 7 or 13. That does not make sense because mean age should be a single number. If you are reporting mean and SD (or mean and SE, or mean and CI) what is presented should be explicit.

• I questioned about presenting multivariable logistic regression results as multivariate logistic regression results. Authors’ response shows they haven’t understood the question. Question was about inappropriate terminology. I know multivariate and multivariable are interchangeably used in literature to mean the same think, but that is erroneous. I invite authors to know the difference between those two very different types of models, and not to present multivariable models as multivariate models.

• Family history as a possible factor. Authors’ response was exactly what I assumed. This response needs to be in the manuscript to avoid the need for readers also to assume the same. I think the use of "family history" is a misleading term here.

• table3. died=48. Authors’ response needs to be included in the manuscript for readers.
---

## [Author Response · Author response to Decision Letter 1]

13 Feb 2023

Response to reviewers comments have addressed in response to reviewer file. Authors haven't cited any retracted article in their knowledge.

---

## [Editor Report · Decision Letter 2]

17 Feb 2023

PONE-D-22-21686R2Unsuccessful treatment outcomes and associated risk factors: A prospective study of DR-TB patients from a high burden country, PakistanPLOS ONE

Dear Dr. Massud,

Thank you for submitting your manuscript to PLOS ONE. After careful consideration, we feel that it has merit but does not fully meet PLOS ONE’s publication criteria as it currently stands. Therefore, we invite you to submit a revised version of the manuscript that addresses the points raised during the review process.

We look forward to receiving your revised manuscript.

Kind regards,

Ari Samaranayaka, PhD

Academic Editor

PLOS ONE

Journal Requirements:

Additional Editor Comments:

Thanks for the revision that attended most of the comments made before. Results and discussion sections were not properly assessed in previous reviews because results are not meaningful until methodology used for obtaining those results are clear enough. I want authors' attention to below points in order it to be at acceptable level.

1. Table4. I had to spent some time to decide what it is first column. I realised in includes the number of people in the category with with unsuccessful outcomes and that number as a percentage of the total number in the category (as opposed to total number in the category and of them % with unsuccessful outcomes). However, some of the presented univariable ORs are inconsistent with that. Eg. Residence variable. Authors need to check presented numbers.

2. Table4 . Age variable. What is the meaning of 45/9% for >50 aged? Why age=50 people were not included? I noted they are included in table5.

3. Table4 . No one was resistant to Eto. Did your SPSS software allowed such a variable in logistic regression to derive Pvalue?

4. All tables, make sure number of decimal places are consistent.

5. Table5. How do you interpret the Pvalue reported for reference group in Baseline Sputum grading variable?

---

## [Author Response · Author response to Decision Letter 2]

15 Apr 2023

All the comments have been addressed.

---

## [Editor Report · Decision Letter 3]

18 Apr 2023

PONE-D-22-21686R3Unsuccessful treatment outcomes and associated risk factors: A prospective study of DR-TB patients from a high burden country, PakistanPLOS ONE

Dear Dr. Massud,

Thank you for submitting your manuscript to PLOS ONE. After careful consideration, we feel that it has merit but does not fully meet PLOS ONE’s publication criteria as it currently stands. Therefore, we invite you to submit a revised version of the manuscript that addresses the points raised during the review process.

We look forward to receiving your revised manuscript.

Kind regards,

Ari Samaranayaka, PhD

Academic Editor

PLOS ONE

Journal Requirements:

Additional Editor Comments:

In last review I asked the meaning of Pvalue presented for the reference category in Baseline Sputum grading variable in table5. As the response to that, reference category has been removed from the table. Other 2 categories of that variable remain with their Odds ratios. That makes unable to interpret those two odds ratios. Whenever category-specific odds ratios are presented for a multi-category variable, each odds ratio is interpreted relative to the reference category, therefore reference category should be explicitly mentioned, unless intuitive in the context. Can authors attend to that please. Otherwise the manuscript is very close to acceptance.

---

## [Author Response · Author response to Decision Letter 3]

15 Jun 2023

The response to reviewers comments have been presented in a separate file.

---

## [Editor Report · Decision Letter 4]

19 Jun 2023

Unsuccessful treatment outcomes and associated risk factors: A prospective study of DR-TB patients from a high burden country, Pakistan

PONE-D-22-21686R4

Dear Dr. Massud,

We’re pleased to inform you that your manuscript has been judged scientifically suitable for publication and will be formally accepted for publication once it meets all outstanding technical requirements.

Kind regards,

Ari Samaranayaka, PhD

Academic Editor

PLOS ONE
---

## [Editor Report · Acceptance letter]

1 Aug 2023

PONE-D-22-21686R4 

Unsuccessful treatment outcome and associated risk factors. A prospective study of DR-TB patients from a high burden country, Pakistan 

Dear Dr. Massud:

I'm pleased to inform you that your manuscript has been deemed suitable for publication in PLOS ONE. Congratulations! Your manuscript is now with our production department. 

Kind regards, 

on behalf of

Dr. Ari Samaranayaka 

Academic Editor

PLOS ONE